



# A simple model for faceted topographies at normal faults based on an extended stream-power law

Stefan Hergarten[1]

[1]Institut für Geo- und Umweltnaturwissenschaften, Albert-Ludwigs-Universität Freiburg, Albertstr. 23B, 79104 Freiburg, Germany

**Correspondence:** Stefan Hergarten
(stefan.hergarten@geologie.uni-freiburg.de)

**Abstract.** Faceted topographies at normal faults have been studied for more than a century. Since the dip angle of the facets is typically much lower than the dip angle of the fault, it is clear that the facets are not just the exhumed footwall, but have been eroded considerably. It has also been shown that a constant erosion rate in combination with a constant rate of displacement can explain the occurrence of planar facets. Quantitatively, however, the formation of faceted topographies is still not fully understood. In this study, the shared stream-power model for fluvial erosion and sediment transport is used in combination with a recently published extension for hillslopes. As a major theoretical result, it is found that the ratio of the tangent of the facet angle and the dip angle of the fault as well as the ratio of baseline length and horizontal width of perfect triangular facets mainly depends on the ratio of the horizontal rate of displacement and the hillslope erodibility. Numerical simulations reveal that horizontal displacement is crucial for the formation of triangular facets. For vertical faults, facets are rather polygonal and much longer than wide. While the sizes of individual facets vary strongly, the average size is controlled by the ratio of hillslope erodibility and fluvial erodibility.

## 1 Introduction

Faceted topographies at normal faults are the perhaps most impressing footprints of active tectonics in geomorphology. Although studied for more than a century, the conditions under which they form and what their properties tell about the tectonic conditions are still not fully understood. Tucker et al. (2020) provided several examples of faceted topographies as well as a short history of research on this topic with numerous references to the original literature. In a nutshell, the earliest studies interpreted facets as exhumed fault planes with little modification by erosion. It was, however, already recognized almost 100 years ago that the faceted surfaces are typically less steep than the respective fault planes. While normal faults typically dip at angles between $50°$ and $70°$, faceted surfaces are rarely steeper than $40°$. So faceted surfaces must have been strongly affected by erosion, despite their often strikingly planar shape.

The importance of erosion for the formation of faceted surfaces raises several questions such as:

1. Under which conditions do planar slopes occur?





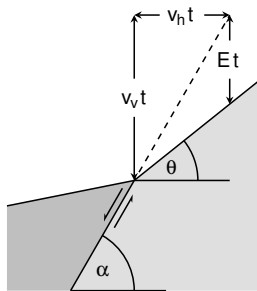

**Figure 1.** Illustration of a planar erosional surface for constant rates of displacement ($v_\mathrm{h}$ and $v_\mathrm{v}$) and erosion ($E$), where $t$ is the time span of exposure.

2. What causes the observed strong variation in dip angles of facets from less than $20°$ (e.g., Menges, 1990) to more than $40°$ (e.g., Wilkinson et al., 2015)?

3. How does the interplay of fluvial and hillslope processes influence shape and size of individual facets?

The first two questions can be answered to some extent with the help of geometrical considerations by Tucker et al. (2011, 2020). As illustrated in Fig. 1, a spatially and temporally constant erosion rate in combination with a constant rate of displacement at the fault yields a planar surface. If $v_\mathrm{h}$ is the horizontal rate of displacement, $v_\mathrm{v}$ the vertical rate (throw rate), and $E$ the erosion rate (measured vertically, not normal to the surface), the angle $\theta$ of the exhumed and eroded surface is given by the relation

$$\tan\theta = \frac{v_\mathrm{v} - E}{v_\mathrm{h}}. \tag{1}$$

In combination with the dip angle $\alpha$ of the fault, which constrains $v_\mathrm{h}$ and $v_\mathrm{v}$ according to

$$\tan\alpha = \frac{v_\mathrm{v}}{v_\mathrm{h}}, \tag{2}$$

this relation can be written in the form

$$\frac{\tan\theta}{\tan\alpha} = 1 - \frac{E}{v_\mathrm{v}}. \tag{3}$$

Assuming $\alpha = 60°$ as a typical dip angle for normal faults, typical facet angles $\theta \leq 40°$ require that the erosion rate is more than half of the fault throw rate $v_\mathrm{v}$ according to Eq. (3). In turn, rather low facet angles of $\theta \leq 20°$ are achieved for $E \geq 0.8 v_\mathrm{v}$. So the simple model with constant rates of displacement and erosion does not only explain the occurrence of planar surfaces qualitatively, but also a large variation in slope for moderate changes in rates of erosion or displacement.

With regard to the third question, drainage patterns upstream of faults have been investigated numerically for more than 15 years (Castelltort and Simpson, 2006; Perron et al., 2008). However, these early studies focused on the apparently regular river spacing (Hovius, 1996; Perron et al., 2009), but not on the hillslopes facing towards the fault.

The number of studies addressing the properties of faceted topographies with the help of numerical landform evolution models still seems to be small. Petit et al. (2009) combined a model of fluvial erosion and sediment transport with a model for





hillslope processes based on a diffusion equation. As a central point, the influence of the climatic (precipitation rate, sediment
transport length, and diffusivity) and tectonic (fault dip angle and slip rate) parameters was investigated systematically. In turn,
however, the study was limited by considering only small domains ($8 \, \text{km} \times 4 \, \text{km}$) with a rather coarse grid (0.1 km spacing).

As a second limitation, the applicability of the diffusion model to hillslope processes may be questioned, although widely
used in large-scale landform evolution modeling. In its simplest form with constant diffusivity $D$, the diffusion equation
predicts convex topographies under erosion. Convex profiles indeed occur frequently on the upper parts of hillslopes in reality
(e.g., Selby, 1985). Petit et al. (2009) already admitted that diffusion is too weak with typical diffusivities obtained from
field observations (e.g., Fernandes and Dietrich, 1997). The dissection of the topography by rivers is too strong then and
the hillslopes are too small. Realistic hillslope sizes can only be achieved by increasing the diffusivity artificially, and the
inherent strong convex curvature can only be avoided by using nonlinear diffusion models with a slope-dependent diffusivity.
Nowadays, the approach suggested by Roering et al. (1999) is widely used, which assumes $D \to \infty$ if the slope approaches a
given limit slope. Petit et al. (2009) assumed that $D$ increases instantaneously by a given factor if the slope exceeds a given
threshold value. Both approaches are typically interpreted as the onset of landsliding and are able to produce planar slopes
under erosion. However, the slope of planar areas is the limit or threshold slope defined in the nonlinear diffusion model then.
This property makes it difficult to explain the occurrence of faceted surfaces with angles lower than typical limits of slope
stability and the observed large variation in dip angles. As a further, rather theoretical argument, combining fluvial erosion
with diffusion typically causes scaling problems (Perron et al., 2008; Pelletier, 2010; Hergarten, 2020a; Hergarten and Pietrek,
2023), which make the results dependent on the spatial resolution of the grid.

The more recent cellular automaton model proposed by Tucker et al. (2020) goes deeper into the hillslope processes. This
model takes into account weathering and partial coverage of the surface by regolith. In turn, however, it is limited to longitudinal
profiles and does not capture the interaction with the drainage pattern.

## 2 Approach and scope

This study is based on the shared stream-power model (Hergarten, 2020b) in combination with the extension for hillslopes
proposed by Hergarten and Pietrek (2023). The shared stream-power model is a simple model that combines fluvial incision
and sediment transport. Mathematically, it is equivalent to the linear decline model (Whipple and Tucker, 2002) and to the $\xi$–$q$
model (Davy and Lague, 2009). The shared stream-power model is described by the equation

$$\frac{E}{K_\text{d}} + \frac{Q}{K_\text{t} A} = A^m S^n, \tag{4}$$

where $E$ is the erosion rate, $Q$ the sediment flux (volume per time), $A$ the upstream catchment size (a proxy for the mean
discharge), and $S$ the channel slope. It involves the 4 parameters $K_\text{d}$, $K_\text{t}$, $m$, and $n$.

The shared stream-power model was designed as an extension of the stream-power incision model (SPIM),

$$E = K A^m S^n, \tag{5}$$



which contains a single erodibility $K$ instead of $K_d$ and $K_t$. For spatially uniform erosion, the sediment flux is $Q = EA$, and Eq. (4) collapses to a form analogous to Eq. (5) with an effective erodibility $K$ according to

$$\frac{1}{K} = \frac{1}{K_d} + \frac{1}{K_t}. \tag{6}$$

So the exponents $m$ and $n$ are the same as in the SPIM. The ratio of $m$ and $n$ is constrained quite well by long profiles of real-world rivers, whereby either $\frac{m}{n} = 0.45$ or $\frac{m}{n} = 0.5$ is typically used (e.g., Whipple et al., 2013; Lague, 2014). The absolute

values of $m$ and $n$ are, however, more uncertain (e.g., Lague, 2014; Harel et al., 2016; Hilley et al., 2019; Adams et al., 2020). The widely used choice $n = 1$ is mainly a matter of convenience since the model is linear with regard to the channel slope $S$ (and thus also with regard to the surface elevation) then.

The parameter $K_d$ describes the ability to erode the riverbed, while the transport capacity

$$Q_c = K_t A^{m+1} S^n \tag{7}$$

(the sediment flux at $E = 0$) is proportional to $K_t$. According to Eq. (6), estimates of $K$ made for the SPIM can be used to constrain $K_d$ and $K_t$, but one degree of freedom remains. The end-members are defined by $K_d = K$ and $K_t \to \infty$, which leads back to the SPIM directly, and the transport-limited model obtained by setting $K_d = \infty$ and $K_t = K$. The findings of Guerit et al. (2019), however, rather suggest $K_d \approx K_t$, which is somehow the middle between the two end-members.

While most of the following considerations could also be performed with the SPIM instead of the shared stream-power

model, the extension for hillslopes proposed by Hergarten and Pietrek (2023) plays a central part. This extension subdivides the model domain dynamically into channel sites and hillslope sites. Flow routing is based on the widely used D8 scheme (O'Callaghan and Mark, 1984) on a regular mesh, which assumes that the entire flow of water and sediment of a site is directed towards the neighbor with the steepest descent. Sites that have only one neighbor with a lower elevation than the site itself are considered channel sites. These are the sites for which a splitting of the fluxes towards multiple neighbors would not make

sense. Other sites are considered hillslope sites. As an additional rule for delineating channels, it was assumed that the flow target of a channel site is also a channel site under all conditions.

Hergarten and Pietrek (2023) proposed basically the same shared stream-power model for hillslopes as for channels (Eq. 4), but with $m = 0$,

$$\frac{E}{\kappa_d} + \frac{Q}{\kappa_t A} = S^n. \tag{8}$$

The erodibilities $\kappa_d$ and $\kappa_t$ (originally termed $\tilde{K}_d$ and $\tilde{K}_t$) are different from $K_d$ and $K_t$ and also have different physical dimensions. In principle, even the exponent $n$ may differ between channels and hillslopes, which is not taken into account in the following.

As a main result, a self-organization of channels and hillslopes was found. While channel formation does not take place at a unique catchment size, the observed range of channel-forming catchment sizes depends on the values of $\kappa_d$ and $\kappa_t$ in relation

to $K_d$ and $K_t$. As an advantage over models based on a diffusion equation, the extended stream-power approach appears to be free of scaling problems, which ensures that numerical simulations performed on grids with different spatial resolutions are consistent.





Concerning the scope of this study, it should be kept in mind that assuming $m = 0$ for hillslopes already defines planar hillslopes as a preferred morphology. For spatially uniform erosion, Eq. (8) turns into

$$E = \kappa S^n \tag{9}$$

with

$$\frac{1}{\kappa} = \frac{1}{\kappa_\mathrm{d}} + \frac{1}{\kappa_\mathrm{t}}, \tag{10}$$

and thus

$$S = \left(\frac{E}{\kappa}\right)^{\frac{1}{n}} \tag{11}$$

Therefore, $S$ is constant for spatially uniform erosion, So straight hillslope profiles are some kind of preferred state in this model, in contrast to concave river profiles (for $m > 0$) or convex hillslopes predicted by the diffusion equation.

Since straight hillslopes are already a part of the model concept, this study cannot address the question why faceted topographies typically have little curvature. Instead, the key points are the geometrical properties (length, width, slope) of facets at normal faults and how they differ from hillslopes along rivers. Simple estimates for triangular facets in equilibrium will be derived analytically in the following section. In Sect. 5, numerical simulations will be performed in order to find out why normal faults facilitate the formation of triangular facets compared to vertical faults and how the drainage network affects the evolution.

## 3   Theoretical considerations

Let us start by considering a river that crosses a normal fault (Fig. 2). If we assign the coordinate system to the footwall, the hanging wall moves at a velocity $v_\mathrm{h}$ horizontally (to the left) and at a velocity $v_\mathrm{v}$ vertically (downward). In the simplest scenario, the river erodes at a uniform erosion rate $E$ into the footwall and neither erodes the hanging wall nor deposits sediments there. Then the erosion rate at the footwall can be described by Eq. (5) with the effective erodibility $K$ defined in Eq. (6).

Equation (1) can be transferred to this scenario by replacing $\tan\theta$ with the channel slope $S = -\frac{\partial H}{\partial x}$. In combination with Eq. (5), we obtain

$$KA^m S^n = v_\mathrm{v} - S v_\mathrm{h}, \tag{12}$$

and for the linear version of the sheared stream-power model ($n = 1$)

$$S = \frac{v_\mathrm{v}}{KA^m + v_\mathrm{h}}. \tag{13}$$

Strictly speaking, the solution with a uniform erosion rate cannot exist since the channel slope decreases downstream at uniform erosion. However, $S$ is typically much smaller than the tangent of the fault angle (Eq. 2), so the variation in erosion rate (and its difference towards $v_\mathrm{v}$) is small.





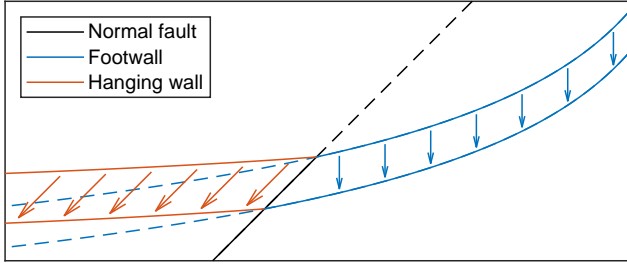

**Figure 2.** Long profile of a river crossing a normal fault. Solid blue lines describe the river incising into the footwall at a constant rate. Dashed blue lines are the continuation of the profile into the hanging wall. Red lines describe the river at the hanging wall, which is just following the movement of the hanging wall.

So horizontal displacement reduces the erosion rate and thus also the equilibrium channel slope at the footwall compared to a vertical fault. For large rivers, however, the effect is weak. If we assume a typical erodibility of $K = 2.5\ \mathrm{Myr}^{-1}$ (e.g., Robl et al., 2017) for $m = 0.5$ and $n = 1$, horizontal movement with $v_{\mathrm{h}} = 1\ \mathrm{mm\,yr}^{-1}$ reduces the channel slope only by 4 % for $A = 100\ \mathrm{km}^2$.

For hillslopes, however, the effect of horizontal movement is stronger. The respective relations are readily obtained by replacing $K A^m$ with $\kappa$ in Eqs. (12), and (13), which yields the relation

$$\kappa S_{\mathrm{f}}^n = v_{\mathrm{v}} - S_{\mathrm{f}} v_{\mathrm{h}}. \tag{14}$$

for the facet slope $S_{\mathrm{f}}$, and for $n = 1$

$$S_{\mathrm{f}} = \frac{v_{\mathrm{v}}}{\kappa + v_{\mathrm{h}}}. \tag{15}$$

This relation can be written conveniently in terms of the dip angles $\alpha$ of the fault (Eq. 2) and of the facet ($\tan\theta = S_{\mathrm{f}}$),

$$\frac{\tan\theta}{\tan\alpha} = \frac{1}{1 + \frac{\kappa}{v_{\mathrm{h}}}}. \tag{16}$$

Figure 3 shows the obtained facet angle $\theta$ as a function of $\frac{v_{\mathrm{h}}}{\kappa}$. The ratio $\frac{v_{\mathrm{h}}}{\kappa}$ has a strong influence on $\theta$ for $n = 1$. A variation from $\frac{v_{\mathrm{h}}}{\kappa} = 0.25$ to 1 covers the range from $\theta = 20°$ to $40°$ for $\alpha = 60°$. We will see in Sect. 5 that $\kappa = 0.75\ \mathrm{mm\,yr}^{-1}$ is a reasonable value. Then the respective range in total displacement $v_{\mathrm{h}}$ is from $0.19\ \mathrm{mm\,yr}^{-1}$ to $0.75\ \mathrm{mm\,yr}^{-1}$, equivalent to a total rate of displacement from $0.38\ \mathrm{mm\,yr}^{-1}$ to $1.5\ \mathrm{mm\,yr}^{-1}$. The dependence of $\theta$ on $\frac{v_{\mathrm{h}}}{\kappa}$ becomes weaker for $n > 1$. This means that we need a greater range in $v_{\mathrm{h}}$ to explain a given variation in $\theta$ at constant $\alpha$ than for $n = 1$.

The length-to-width ratio of perfect triangular facets can also be determined analytically. Let us assume that the facet is bounded by two rivers flowing perpendicularly to the fault (Fig. 4) with identical catchment sizes large enough to ensure that the effect of horizontal movement on their erosion rate is negligible. Then their erosion rate is $E = v_{\mathrm{v}}$ and their channel slope is

$$S = \left(\frac{v_{\mathrm{v}}}{K A^m}\right)^{\frac{1}{n}}. \tag{17}$$





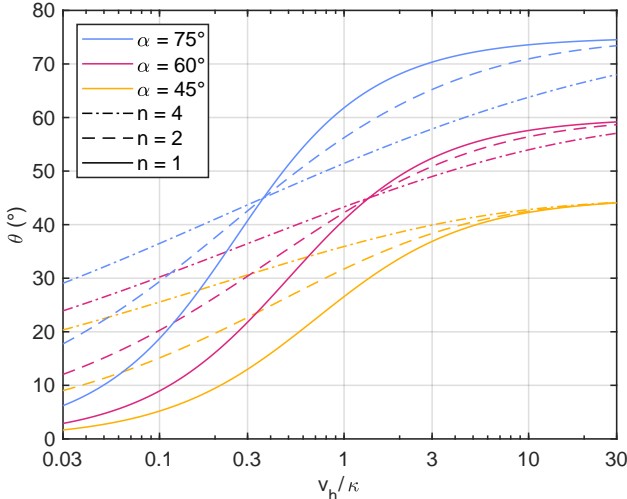

**Figure 3.** Facet angle as a function of the ratio $\frac{v_h}{\kappa}$ (Eq. 14 with $\tan\theta = S_f$) for different fault angles $\alpha$ and different values of the exponent $n$.

Since the erosion rate of the hillslopes draining into these rivers must be the same, the respective slope of the hillslopes must be

$$S_h = \left(\frac{v_v}{\kappa}\right)^{\frac{1}{n}}. \tag{18}$$

If $A$ is sufficiently large, the direction of steepest descent at the hillslopes is normal to the rivers. Then the height of the triple junction of the facet and the two hillslopes must be

$$h = S_f w = Sw + S_h\frac{b}{2}, \tag{19}$$

with the baseline length $b$ and the width $w$ measured horizontally (Fig. 4). Equation (19) yields the length-to-width ratio

$$\frac{b}{w} = 2\frac{S_f - S}{S_h} \tag{20}$$

with $S$ from Eq. (17), $S_h$ from Eq. (18), and $S_f$ defined implicitly by Eq. (14). Using Eq. (15), this ratio can be computed analytically for $n = 1$,

$$\frac{b}{w} = 2\left(\frac{\kappa}{\kappa + v_h} - \frac{\kappa}{KA^m}\right). \tag{21}$$

The second term in Eq. (21) can be neglected if the rivers are sufficiently large ($KA^m \gg \kappa$). Then the length-to-width ratio only depends on the ratio $\frac{v_h}{\kappa}$ in the form

$$\frac{b}{w} = \frac{2}{1 + \frac{v_h}{\kappa}}. \tag{22}$$





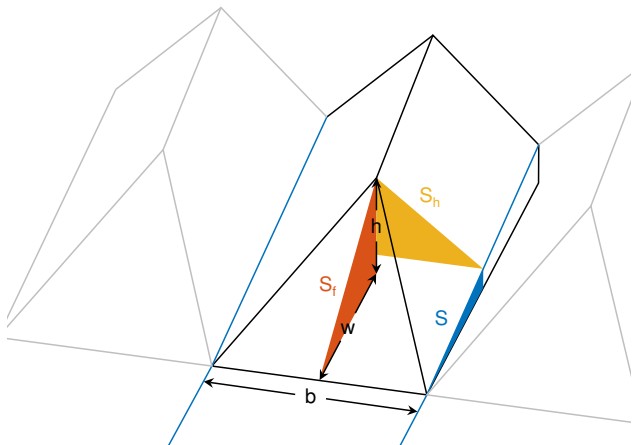

**Figure 4.** Geometry of triangular facets between parallel rivers.

In this case, the ratio $\frac{v_\mathrm{h}}{\kappa}$ controls the ratio of the tangents of the dip angles (Eq. 16) as well as the shape of perfect triangular facets (Eq. 22).

If the rivers are small, the ratio $\frac{\kappa}{KA^m}$ comes in as a second control. This ratio describes erosion at hillslopes relative to erosion in channels. An increase in this ratio makes the facets wider in relation to their length.

Equation (21) can also be generalized to facets bounded by rivers with different catchment sizes. All considerations have to be made for the left-hand part and the right-hand part of the facet separately, where $\frac{b}{2}$ has to be replaced by the width of the respective part. Adding the two widths yields the relation

$$\frac{b}{w} = 2\frac{\kappa}{\kappa + v_\mathrm{h}} - \frac{\kappa}{KA_1^m} - \frac{\kappa}{KA_2^m} \tag{23}$$

instead of Eq. (21), where $A_1$ and $A_2$ are the catchment sizes of the two rivers. As a major difference, the facets become
asymmetric.

## 4  Validation

As mentioned in Sect. 2, there is still uncertainty about the value of the exponent $n$ in stream-power erosion models. This uncertainty is even higher for the extension towards hillslopes since it is not clear whether the value of $n$ is the same as for rivers. As a first validation of the approach and in order to get an estimate of $n$, the geometrical properties of facets at 10 normal
faults in Greece and Bulgaria published by Tsimi and Ganas (2015) are used. The data listed in Table 1 consist of the mean values for each fault, taken over 13 to 66 facets.

For each fault, the mean dip angle $\alpha$ is used as a given property. If a range is given, the mean value is used. The ratio $\frac{v_\mathrm{v}}{\kappa}$ is then adjusted for each fault individually to fit the slopes of the fault $S_\mathrm{f} = \tan\theta$ (Eq. 14 with $v_\mathrm{h} = \frac{v_\mathrm{v}}{\tan\alpha}$) and the hillslopes $S_\mathrm{h} = 2\frac{h}{b}$ (Eq. 18, assuming $S \ll S_\mathrm{h}$). The deviations obtained for the 10 faults are shown in Fig. 5.



**Table 1.** Average data from 10 faults investigated by Tsimi and Ganas (2015). The last column contains the best-fit value of $\kappa$ for $n = 1$.

| No. | $\alpha$ (°) | $\theta$ (°) | $\frac{h}{b}$ | $v_\mathrm{v}$ (mm yr$^{-1}$) | $\kappa$ (mm yr$^{-1}$) |
|---|---|---|---|---|---|
| 1 | 47 | 26.13 | 0.40 | 0.29 | 0.36 |
| 2 | 50–60 | 25.61 | 0.5 | 0.29 | 0.30 |
| 3 | 65 | 24.13 | 0.38 | 0.50 | 0.70 |
| 4 | 50 | 36.80 | 0.97 | 0.84 | 0.43 |
| 5 | 70 | 32.18 | 0.67 | 0.75 | 0.61 |
| 6 | 50–60 | 28.39 | 0.58 | 0.29 | 0.26 |
| 7 | 38–52 | 22.79 | 0.43 | 0.21 | 0.25 |
| 8 | 55 | 29.32 | 0.52 | 0.74 | 0.72 |
| 9 | 38–80 | 31.86 | 0.64 | 0.74 | 0.59 |
| 10 | 65–85 | 27.67 | 0.44 | 0.66 | 0.83 |

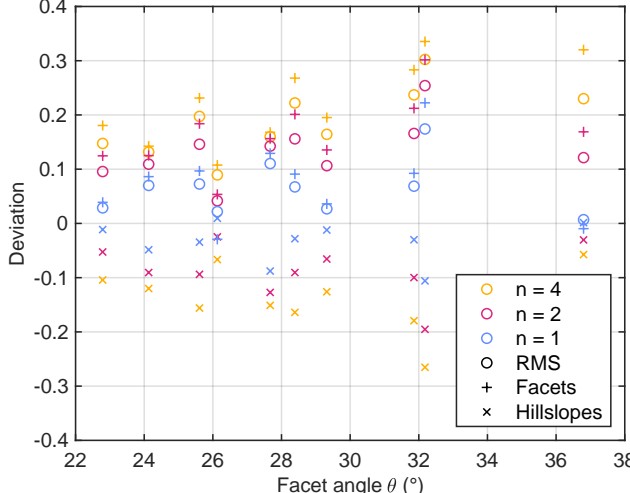

**Figure 5.** Deviations in $S_\mathrm{f}$ (facets) and $S_\mathrm{h}$ (hillslopes) obtained by fitting the ratio $\frac{v_\mathrm{v}}{\kappa}$ for each of the faults considered by Tsimi and Ganas (2015). The root-mean-square (RMS) deviation is computed from the deviations in $S_\mathrm{f}$ and $S_\mathrm{h}$ and is always positive, while the individual deviations may be positive or negative.

For all faults, the deviations for $n = 1$ are lower than for $n = 2$ and for $n = 4$. This holds for the root-mean-square (RMS) deviation as well as for the individual deviations in $S_\mathrm{f}$ and $S_\mathrm{h}$. The deviations are smaller than 0.1 for 8 out of the 10 faults for $n = 1$. The obtained $R^2$ values range from 0.36 for $n = 4$ to 0.89 for $n = 1$.

The model systematically overestimates $S_\mathrm{f}$ and underestimates $S_\mathrm{h}$ and thus underestimates the difference between both, except for 2 faults at $n = 1$. These findings suggest that the best-fit exponent may even be $n < 1$ for the data considered here.

However, the data set is limited and should not be overrated. Nevertheless, the results suggest that $n = 1$, which makes the model linear, is a reasonable choice for hillslopes rather than any value $n > 1$.

After fitting the $\frac{v_v}{\kappa}$ for each fault, the respective hillslope erodibility $\kappa$ can be estimated using the given throw rate $v_v$. The obtained values cover a range from $\kappa = 0.25\ \mathrm{mm\,yr^{-1}}$ to $0.83\ \mathrm{mm\,yr^{-1}}$ (Table 1) with a mean value of $0.51\ \mathrm{mm\,yr^{-1}}$.

## 5 Numerical simulations

While the theoretical considerations from Sect. 3 predict some geometrical properties of faceted topographies, the role of the drainage network has to be explored numerically. In this section, the formation of triangular and general polygonal facets, facet sizes, and the development of faceted topographies through time will be investigated.

### 5.1 Numerical setup

All simulations are based on the the shared stream-power model with the extension for hillslopes described in Sect. 2 and the simplest parameter choice $m = 0.5$ and $n = 1$. As discussed by Hergarten and Pietrek (2023), converting nondimensional coordinates to real-world properties is particularly simple then since time, horizontal length scale, and height scale are completely independent.

All simulations were performed with the landform evolution model OpenLEM (Hergarten, 2024a) on a regular $6000 \times 6000$ grid. The southern and northern boundary are kept at zero elevation and periodic boundary conditions are applied to the eastern and western boundaries. The only real-world length scale already included in the model design is that the grid spacing should be $\delta x = 10\ \mathrm{m}$. This is a quite high resolution compared to typical applications of large-scale landform evolution model, which was chosen in order to allow for simulating a large number of facets with a reasonable resolution, given that typical facet sizes are some hundred meters.

The time scale and the height scale are in principle arbitrary and all results can be rescaled afterwards. As the dimension of the erodibilities is inverse time, the time scale is immediately defined by the real-world values of the erodibilities. Since the findings of Guerit et al. (2019) suggest $K_d \approx K_t$, $K_d = K_t = 2$ is a convenient choice in nondimensional coordinates. Then the effective erodibility is $K = 1$ according to Eq. (6). If we define, e.g., $K = 2.5\ \mathrm{Myr^{-1}}$ (Robl et al., 2017) as a real-world erodibility, one nondimensional time unit corresponds to a time span of $T = \frac{1}{K} = 0.4\ \mathrm{Myr}$. This time scale is used for illustration in the following, keeping in mind that it could be adjusted easily. A vertical length scale of $2\ \mathrm{m}$ is used for convenience, which could also be adjusted easily.

An equilibrium topography with a uniform uplift rate $U = 1$ ($2\ \mathrm{m}$ per $0.4\ \mathrm{Myr}$ with the scaling defined above) is starting point of all simulations. The respective topography is shown in Fig. 6. As discussed by Hergarten and Pietrek (2023), the model with the hillslope extension does not yield state-state topographies in the strict sense. Using a time increment $\delta t = 10^{-3}$, there are still about 133,000 changes in flow direction per time step (about 0.4 % of all sites), but the maximum change in elevation is constantly below $10^{-15}$, so within the numerical precision. Overall, the relief of the equilibrium topography is small, which means that it is mainly a prescribed drainage pattern on an almost flat topography.



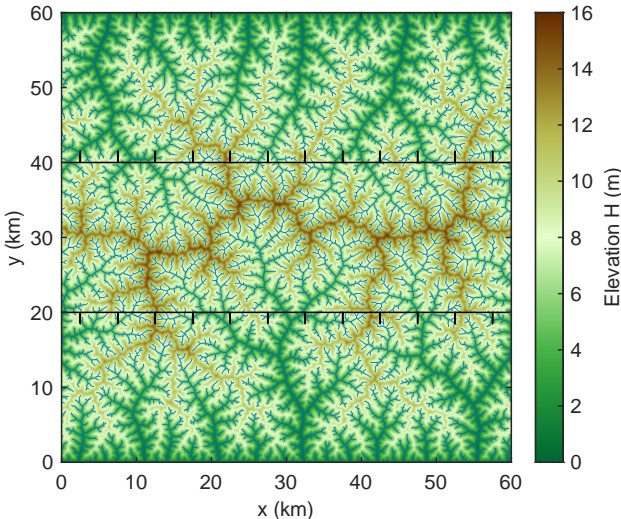

**Figure 6.** Initial equilibrium topography with a horst structure bounded by two normal faults.

The hillslope erodibilities $\kappa_\mathrm{d}$ and $\kappa_\mathrm{t}$ are the only nontrivial parameter choices. For $m = 0.5$, their dimension is length per time. So the ratio of the effective erodibilities $l = \frac{\kappa}{K}$ (Eqs. 6 and 10) defines a length scale, which controls channelization. In this study, $l = 300\,\mathrm{m}$ (30 pixels) is assumed. This choice yields a drainage density (total channel length per area) of $1.09\,\mathrm{km}^{-1}$.

For simplicity, we assume $\frac{\kappa_\mathrm{d}}{K_\mathrm{d}} = \frac{\kappa_\mathrm{t}}{K_\mathrm{t}} = l$.

A horst structure bounded by two normal faults is considered in all simulations (Fig. 6). The two faults are located at $y = 20\,\mathrm{km}$ and $y = 40\,\mathrm{km}$ for $H = 0$, and rates of displacement $v_\mathrm{v}$ and $v_\mathrm{h}$ define the inclination of the fault plane. While the entire domain (including the fault plane) is still uplifted at the low rate $U$, all material above the fault plane (including the boundaries) is continuously lowered at the throw rate $v_\mathrm{v}$ and moved towards the boundaries at the velocity $v_\mathrm{h}$.

Since the model OpenLEM assumes a fixed regular grid, displacement is performed in discrete steps of one mesh width. This means that all material above the fault plain is moved towards the boundaries after time intervals of $\tau = \frac{\delta x}{v_\mathrm{v}}$. In order to reduce potential artifacts arising from the discrete steps, data are only evaluated at times in the middle between steps of displacement.

Due to the linearity of the model for $n = 1$, $v_\mathrm{v}$ only defines how steep the topography will become, but has no further effect, provided that it is much higher than $U$. A throw rate of $v_\mathrm{v} = 100$ ($0.5\,\mathrm{mm\,yr}^{-1}$ with the scaling defined above) is assumed in

all simulations, while the effect of $v_\mathrm{h}$ will be investigated.

In order to keep the analysis simple, facets are defined by their drainage pattern and not by their planar shape at first. In this sense, a facet is a continuous set of hillslope nodes at the footwall that drains directly to the outcrop line of the fault without forming a channel. Since small transient channels form occasionally close to the fault, channels reaching less than 10 pixels ($100\,\mathrm{m}$) into the footwall are disregarded. Formally, facets defined this way are just hillslopes, and their planar shape will be

investigated afterwards.





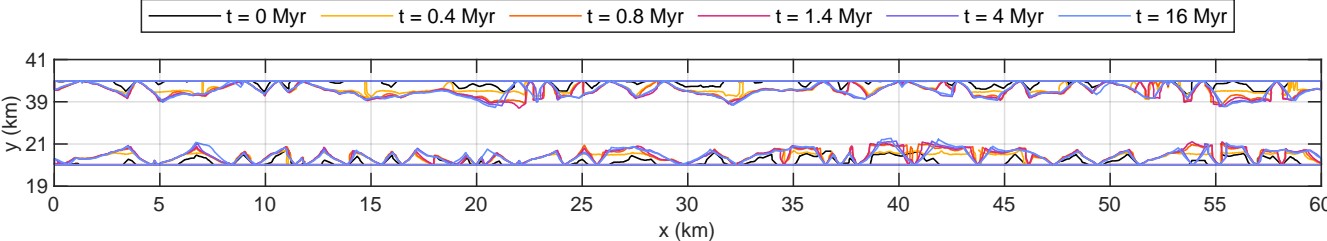

**Figure 7.** Formation of facets at the two vertical faults. Only the part of the domain around the faults is shown.

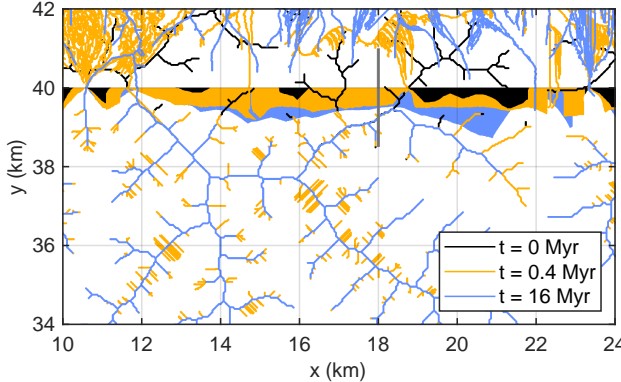

**Figure 8.** Evolution of the channel pattern around the longest facet shown in Fig. 7. The gray line shows the profile analyzed in Fig. 9.

## 5.2 Vertical faults

As a first step towards understanding the formation of faceted topographies, vertical faults are considered. Since there is no horizontal displacement, there is no need to displace the hanging walls in discrete steps as described in the previous section. A continuous vertical displacement at the faults is assumed, which means that the northern and southern parts of the domain

($y < 20$ km and $y > 40$ km) are continuously lowered.

Figure 7 shows the evolution of facets (without regard to their planar shape) at different times. As the most striking result, the evolution is fast in the beginning but ceases soon. While a few triangular facets form, large parts of the faceted area consist of polygonal facets with baseline lengths $b$ of up to several kilometers. Breaking these long facets into smaller, potentially more triangular facets seems to be impossible.

The persistence of these long, polygonal facets is related to the persistence of the drainage pattern. Figure 8 shows the change in drainage pattern around the longest facet. The occurrence of this facet is related to a river flowing almost parallel to the fault for about 10 km. The biggest changes in the drainage pattern occur in the hanging wall. Rivers draining from the hanging wall across the fault are rapidly disconnected. The drainage pattern on the hanging wall, which is dominated by the deposition of sediments, becomes quite irregular and changes rapidly. These changes are, however, not able to modify the drainage pattern

on the footwall strongly.





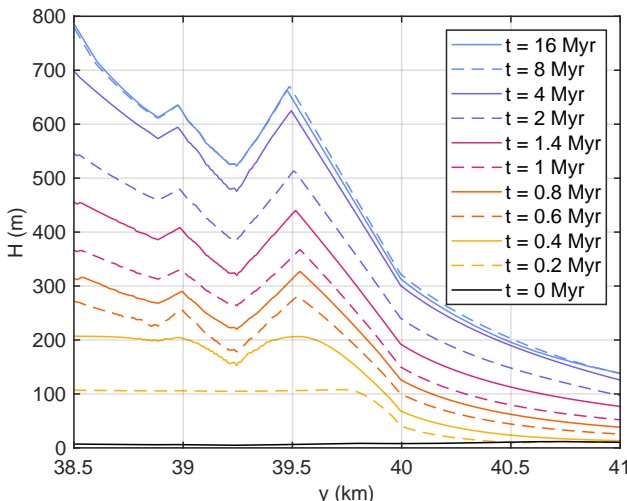

**Figure 9.** Topographic profiles along the gray line shown in Fig. 8. The height $H$ is measured relative to the boundary of the domain.

The effect of the more or less persistent river pattern becomes visible in the north-south profile shown in Fig. 9. At $t = 0.2$ Myr, relief is still small across the section on the footwall. This part of the domain has mainly been uplifted by 100 m without changes in erosion rate. At $t = 0.4$ Myr, however, the river crossing the profile at $y \approx 39.2$ km has already incised into the footwall. This incision finally inhibits the growth of the facet. The hillslopes draining towards this river and towards the

fault become planar (straight in the profile) soon. Since the erosion rate of the river finally equals the fault throw rate, erosion rates are the same at both hillslopes, resulting in identical slope angles. The drainage divide does not move in this situation, which means that the facet is somehow locked and cannot change much.

The retarded incision of the river into the footwall is related to knickpoint migration starting from the fault. As discussed by Hergarten (2021), knickpoints migrate upstream at a speed $K_\mathrm{d} A^m$ in the shared stream-power model for $n = 1$. Similarly, the

speed is $\kappa_\mathrm{d}$ at hillslopes. As long as the rivers are not too small, knickpoint migration in rivers is much faster than on hillslopes. Although the river considered here has a length of more than 7 km from the fault to the profile, the knickpoint arrives earlier than the knickpoint from the fault along the facet would arrive. Otherwise, the river would be deflected towards to fault and the facet would be separated into two facets. This example, however, shows that this process in unlikely, so that the long facets tend to persist.

## 5.3   Normal faults

While knickpoint migration in the rivers at the footwall is basically the same as described in the previous section, the erosion rates at hillslopes facing towards the fault and towards the rivers on the footwall differs. Since the rivers on the footwall define the base level for the respective hillslopes, these hillslopes adopt the erosion rates of the rivers. As found in Sect. 3, this erosion rate is only slightly lower than the throw rate of the fault. In turn, the erosion rate of the facets is considerably lower. As a



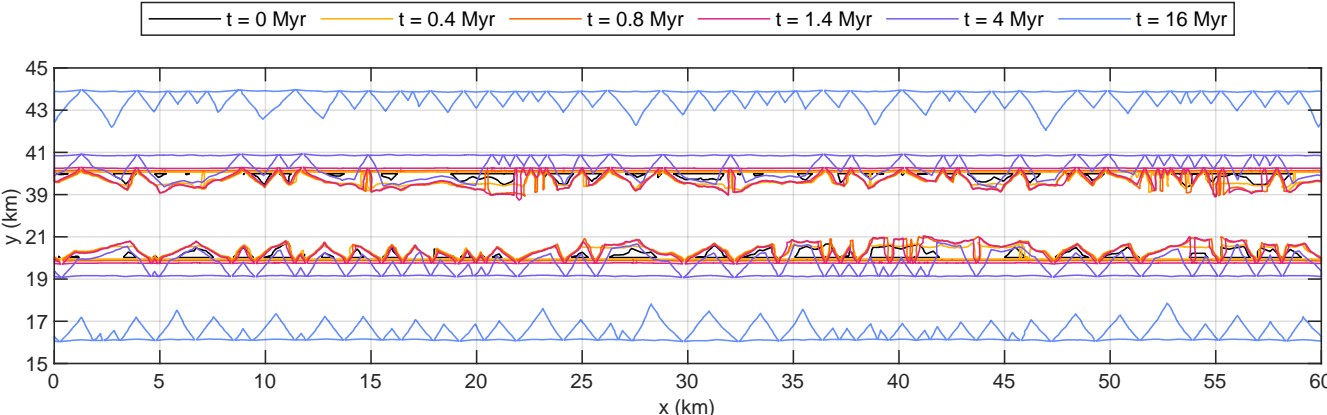

**Figure 10.** Formation of facets at the two normal faults for $v_\text{h} = \frac{1}{3}\kappa$. Only the part of the domain around the faults is shown.

consequence, the drainage divide moves permanently towards the respective fault. Since the outcrop of the fault also migrates, it could be said that the facets slide down the exhumed footwall at the horizontal velocity $v_\text{h}$.

   Figure 10 shows the formation of faceted areas for a moderate rate of horizontal displacement $v_\text{h} = \frac{1}{3}\kappa$. With the scaling defined above, this rate is $v_\text{h} = 0.25 \text{ mm yr}^{-1}$. It it immediately recognized that the original outcrop lines of the faults (at $y = 20 \text{ km}$ and $y = 40 \text{ km}$) play an important part for the evolution of facets. Several large facets still cross the respective line 285    at $t = 4 \text{ Myr}$. While the majority of these facets is polygonal, the parts of their outlines below the original outcrop line are already straight. In turn, all facets are entirely located on the exhumed and eroded fault plane at $t = 16 \text{ Myr}$ and are triangular.

   As shown in Fig. 11, the channel pattern on the exhumed and eroded footwall (between the original and the actual outcrop line) is totally different from that in the rest of the domain. It consists of more or less straight rivers normal to the fault. In contrast, the channel pattern upstream of the original outcrop is almost unaffected, similar to the situation at the vertical fault 290    considered in Sect. 5.2.

   Asymmetric triangular facets are visible at $x \approx 12 \text{ km}$ and at $x \approx 17 \text{ km}$, where the rivers at both sides of the facets differ strongly in length and thus also in catchment size. Returning to Fig. 10, however, it is recognized that strongly asymmetric facets are rare in this scenario.

   The transition from polygonal to triangular facets is also visible in the analysis of the facet sizes shown in Fig. 12. Long 295    facets form rapidly in the beginning, and their width increases through time. Long facets are finally separated into smaller facets, causing an increase in the number of facets between $t = 4 \text{ Myr}$ and $t = 16 \text{ Myr}$. The ratio of baseline length and horizontal width approaches the theoretical relation for large rivers (Eq. 22) during this phase. Mean and median baseline length are about 1.4 km at $t = 16 \text{ Myr}$ with a standard deviation of about 0.6 km.

   Figure 13 shows the formation of faceted areas for a faster horizontal displacement with $v_\text{h} = \kappa$ (0.75 mm yr$^{-1}$ with the 300    scaling defined above). In order not to come too close to the boundary, the last state is $t = 8 \text{ Myr}$ instead of $t = 16 \text{ Myr}$.





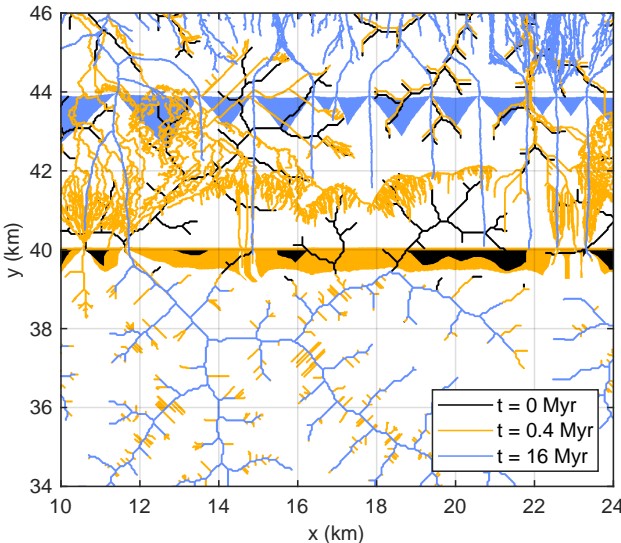

**Figure 11.** Evolution of the channel pattern for $v_h = \frac{1}{3}\kappa$ around the fault segment shown in Fig. 8.

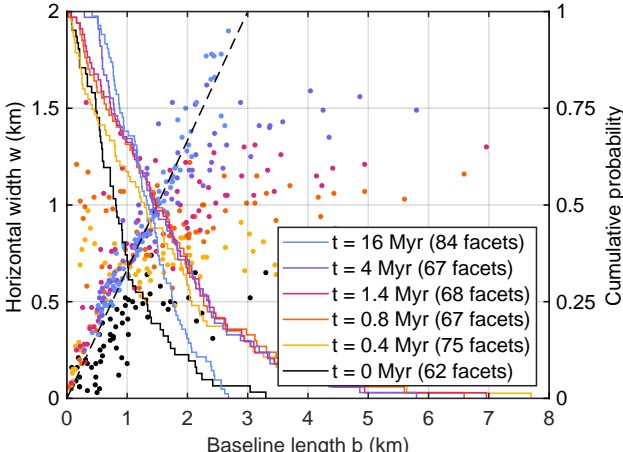

**Figure 12.** Facet sizes for $v_h = \frac{1}{3}\kappa$. Each dot refers to an individual facet, and the dashed line describes the theoretical relation between baseline length and horizontal width for large rivers (Eq. 22). Solid lines show the inverse cumulative probability of the baseline length, i.e., the probability that a randomly picked facet has a baseline length of at least $b$.

The behavior is qualitatively similar to that observed for the lower rate of displacement. Figure 14, however, reveals that the geometry of the facets is different. While Eq. (21) predicts $w = \frac{2}{3}b$ for the previous scenario ($v_h = \frac{1}{3}\kappa$), it predicts $w = b$ here. In turn, the facets are shorter now, with a mean and median baseline length of about $1\,\mathrm{km}$ at $t = 8\,\mathrm{Myr}$. This shortening almost compensates the different length-to-width ratio, so that the horizontal width is about $1\,\mathrm{km}$ on average in both scenarios.





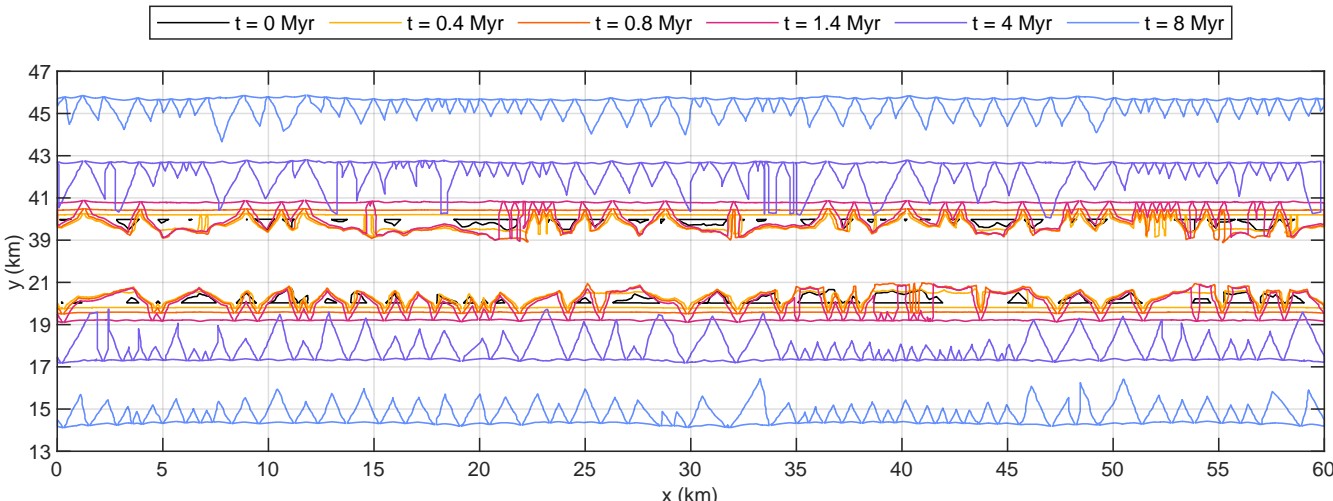

**Figure 13.** Formation of facets at the two normal faults for $v_\mathrm{h} = \kappa$. Only the part of the domain around the faults is shown.

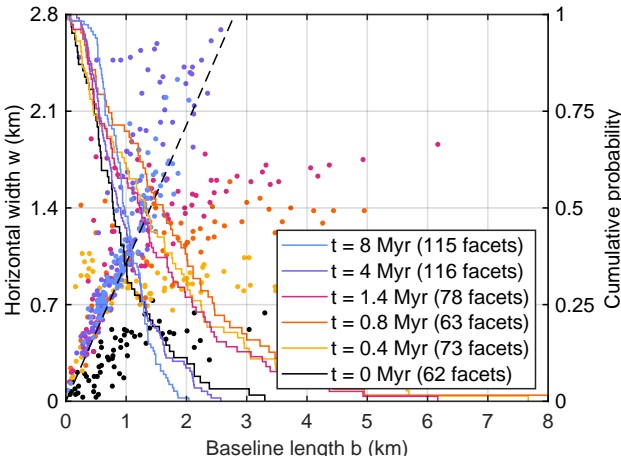

**Figure 14.** Facet sizes for $v_\mathrm{h} = \kappa$. Each dot refers to an individual facet, and the dashed line describes the theoretical relation between baseline length and horizontal width for large rivers (Eq. 21 for $A \to \infty$). Solid lines show the inverse cumulative probability of the baseline length.

As discussed in Sect. 3 (Eq. 23), smaller catchment sizes of the rivers reduce the ratio $\frac{b}{w}$, corresponding to a shift to the left compared to the dashed straight line in Fig. 14. This trend is clearly visible here, while it was not in Fig. 12. The stronger trend for faster horizontal displacement arises from the decrease in the first term in Eq. (23) with increasing $v_\mathrm{h}$, which increases the effect of the finite catchment size relatively. As a further consequence, the tendency towards asymmetric facets becomes stronger, which is visible to some extent in Fig. 14 in comparison to Fig. 12.



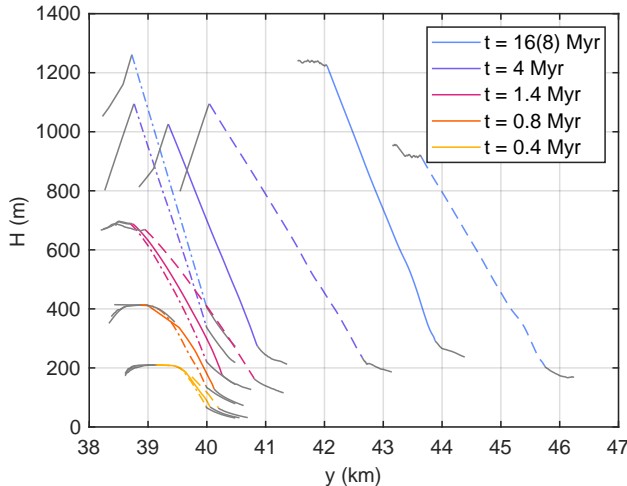

**Figure 15.** North-south profiles across the largest facets at the northern fault, taken along the line of greatest width. Solid lines refer to $v_\mathrm{h} = \frac{1}{3}\kappa$, dashed lines to $v_\mathrm{h} = \kappa$ (with $t = 8$ Myr instead of $t = 16$ Myr), and dash-dotted lines to $v_\mathrm{h} = 0$. For illustration, the profiles are extended by 500 m beyond the facets (gray lines).

Finally, the topography of the faceted areas is investigated in Fig. 15, where a single profile is plotted across the largest facet at the northern fault for each scenario ($v_\mathrm{h} = 0$, $v_\mathrm{h} = \frac{1}{3}\kappa$, and $v_\mathrm{h} = \kappa$) and each time. While the profiles show a convex curvature at early times, they become more or less straight at $t = 4$ Myr for all considered scenarios, indicating a planar shape of the facets. So the transition from convex hillslopes to planar facet appears to be independent of the horizontal displacement rate. In particular, the transition from convex to planar hillslopes appears to be at least as fast as the transition from polygonal to triangular areas.

## 6 Conclusions

In this study, the shared stream-power model was used in combination with a recently published extension for hillslopes (Hergarten and Pietrek, 2023) for modeling the evolution of faceted topographies at normal faults. A first validation based on data published by Tsimi and Ganas (2015) yielded a better fit for the linear version of the model than for nonlinear versions with exponents $n > 1$. However, this result only applies to the exponent in the stream-power model used for the hillslopes, while the exponent may be different in the fluvial regime.

Two theoretical relations for the geometry of perfect triangular facets were derived. The first relation (Eq. 16) refers to the cross-sectional geometry and describes the ratio of the tangent of the facet angle $\theta$ and the dip angle $\alpha$ of the fault. The second relation (Eq. 21 or 23) describes the ratio of baseline length and horizontal width.

As a major result, the nondimensional ratio of the horizontal rate of displacement $v_\mathrm{h}$ and the hillslope erodibility $\kappa$ is the only parameter in both relations. Strictly speaking, this holds for the second relation only in the limit of infinite catchment sizes



of the rivers between the facets (Eq. 22). The facets become relatively wider for finite catchment sizes and become asymmetric if the catchment sizes of the adjacent rivers differ. These effects become stronger with increasing rate of displacement.

The formation of facets and the question whether triangular or polygonal facets are preferred had to be addressed numerically.
The persistence of the drainage network in the region upstream of the fault plays a central part in this context. The limited ability to modify the existing channel network inhibits the expansion of facets into this region. In turn, a new drainage network is created on the exhumed fault plane, allowing for the formation of facets. These facets are shifted downslope through time and typically achieve a triangular shape as soon as they are entirely located on the exhumed fault plane. For this reason, vertical faults produce long and narrow polygonal facets rather than triangular facets.

The size of the facets varies strongly even after they have achieved a triangular shape. While the mean baseline length decreases with increasing horizontal displacement rate, the horizontal width does not change much on average. The absolute size depends on the ratio of hillslope erodibility and fluvial erodibility, which was $\frac{\kappa}{K} = 300$ m in the simulations. This value yielded a drainage density of $1.09$ km$^{-1}$ and a mean horizontal facet width of about $1$ km. These properties can be rescaled by changing the ratio $\frac{\kappa}{K}$. Increasing this ratio by a given factor reduces the drainage density and increases the facet sizes by the 340 same factor.

Overall, the shared stream-power model with the extension for hillslopes appears to be able to explain the formation of faceted topographies and the geometric properties of individual facets reasonably well. I should, however, be kept in mind that the model already includes planar hillslopes as some kind of preferred state, in contrast to convex hillslopes predicted by the widely used diffusion approach.

*Code and data availability.* All codes are available in a Zenodo repository at https://doi.org/10.5281/zenodo.10473156 (Hergarten, 2024b). This repository also contains the data obtained from the numerical simulations. Users who are interested in using the landform evolution model OpenLEM in their own research are advised to download the most recent version from http://hergarten.at/openlem (Hergarten, 2024a). The author is happy to assist interested readers in reproducing the results and performing subsequent research.

*Video supplement.* A video showing the evolution of the topography is available at http://hergarten.at/openlem/facets (Hergarten, 2024a).

*Competing interests.* The author has declared that there are no competing interests.

*Acknowledgements.* This work was funded by the Deutsche Forschungsgemeinschaft (DFG, German Research Foundation) – 432703650.



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
