# Peer review of "A simple model for faceted topographies at normal faults based on an extended stream-power law"

_EGUsphere, 2024_

## Author Response (AR1)

Dear Reviewers,

thank you for your comments. A the most important point for me, the comments of the first reviewer helped me to recognized where the biggest challenges for the readers may be. Unfortunately, I only found the overall assessment of the second reviewer, but no comments. The points addressed in the first report are discussed below, where changes to the manuscript are highlighted in bold letters. Line numbers refer to the version with highlighted changes.

Best regards,

Stefan Hergarten

**Reviewer 1**

*This paper describes the application of an extended, simple stream-power and hillslope erosion model to investigate the evolution and controls on triangular facets at range fronts. The results highlight specific relationships between vertical and horizontal normal fault displacement rates, and the dip angle and dimensions of facets.*

*The paper contains some useful insights and prompted some ideas for future application, so is worthy of publication. However I suggest (extensive) minor but important revisions, because aspects of the presentation make it harder to follow than it needs to be; several assumptions are simply asserted or referred to a recent paper by the same author, without restating the key justification for the assumption in the current paper; and a few existing relationships between ridge relief and drainage incision and drainage spacing would be helpful comparisons. Detailed suggestions and questions below by line number.*

*1. (Abstract) why is it important that they've been studied for a century? This first line would be better as a brief description of what they are and our most general understanding of them. This would help the reader understand the second sentence, which presumes a detailed understanding of the setting and geometry of these features, which many readers will not have.*

Ok, **I added a short description (lines 1–2).**

*11. what does "rather" polygonal mean? Approximately?*

Should mean that there are typically more than three corners. **I replaced it by "multiangular" throughout the manuscript**.

*13. impressing → impressive ?*

**Changed (line 14).**

*13–15. Same issue as abstract: needs to define what facets are, how they relate to terrain and tectonics, and why we care about them, before jumping into details of geometry and research questions. Again, the length of time they've been studied isn't really relevant as motivation for this study.*

**I extended the first paragraph a bit (lines 16–18).**

*To this end, I also suggest making Figure 4 the first figure, and adding to it 2–3 photos of different-looking range fronts with facets, so the reader can see clearly what geometry you're referring to.*

Good idea! **I added one photo (Fig. 1)** since I do not like including photos of features that I have not yet seen in field myself.

*34. Re-cite Tucker here at the end of this paragraph.*

I did not re-cite Tucker here because this paragraph is just the idea of Tucker written in different terms (e.g., vertical erosion rate instead of normal to the surface). Re-citing it explicitly would somehow suggest that the final equation occurs in the paper of Tucker in this form.

*39. Why is the 15 years relevant?*

It is not relevant, but fits well into the sentence. I do not know why it should be a problem to mention it.

*42–43. This omits the very relevant work of Densmore et al from the late 1990s who numerically modeled these in the Basin and Range (US) and explicitly discussed the planarity and other geometric aspects of the facets.*
*- Ellis, Densmore, and Anderson, 1999, Development of mountainous topography in the Basin Ranges, USA: Basin Research, v. 11, p. 2141, doi:10.1046/j.1365-2117.1999.00087.x.*
*- Densmore, A.L., Ellis, M.A., and Anderson, R.S., 1998, Landsliding and the evolution of normal-faultbounded mountains: Journal of Geophysical Research: Solid Earth (19782012), v. 103, p. 1520315219, doi:10.1029/98jb00510.*

**I included these studies (lines 47–77),** which I saw mainly in the context of landsliding before.

*54–58. Regarding nonlinear hillslope transport, I don't think the slope needs to be as steep as the threshold for rock stability; it includes shallow slips in regolith, nonlocal transport e.g. rocks going all the way down the hill, etc., and so planarity of slopes is approached well before the ultimate threshold slope is reached.*

Correct in reality, but the respective models rather enforce the threshold. The model of Densmore et al. introduces a minimum slope angle below which slopes always remain stable. In turn, the nonlinear diffusion model proposed by Roering at al. introduces a maximum slope that can never be exceeded. **I tried to find a less strict wording (lines 75–76).**

*57. delete "then" at the end of the sentence*

**It has vanished in the rewritten section.**

*59–62. This doesn't follow the discussion earlier in the paragraph, ad could be relocated down to the model description ∼ line 105.*

**It is now part of the motivation behind the model (lines 118–123).**

*98. Justification for $m = 0$?*

**I added a paragraph about the motivation behind the model (lines 118–123),** which hopefully explains why assuming $m = 0$ makes sense.

*115. "Some kind of" → "a"; or perhaps the "only" preferred state*

**Changed (line 136).**

*108–116. Is there any empirical support for this model?*

Not yet, except for the visual impression that many slopes are more of less planar with a sharp kink between slope and channels at the channel heads. I planned a quantitative investigation of this kink as a student's thesis, but I did not find a student for this topic so far.

*131. "sheared" → "shared"*

**Fixed (line 152),** thanks!

*157. "Must be the same" must it? Fault movement is episodic and generates substantial and persistent transients (e.g. Yanites, B.J., Tucker, G.E., Mueller, K.J. and Chen, Y.G., 2010. How rivers react to large earthquakes: Evidence from central Taiwan. Geology, 38(7), pp.639-642.)*

Of course, there will be knickpoints moving upstream if the displacement is not continuous. However, we are discussing a model with continuous displacement here, and the effect of discontinuous displacement would be much weaker at steep hillslopes than in a river. Starting such a discussion at this point of the manuscript seems to be more distracting than helpful to me.

*Section 4 – I really appreciate the inclusion of empirical constraint on n for this application, rather than simply asserting $n = 1$ as many studies do.*

While I was quite happy with the preliminary validation when I wrote the manuscript, I realized that neglecting the channel slope of the rivers may even cause a systematic bias. I still think it goes into the right direction, but feel that **I had to add a remark as a warning (lines 217–220).**

*195. The data set used here is limited, but surely there are slip rate constraints on many ranges in (for example) the US basin and range, and geometries can be measured from DEMs, if more constraint is helpful.*

It would definitely be helpful. However, the validation is more challenging than I thought when writing the manuscript. All relations include only ratios of slip rates and erodibilities, and estimating erodibilities involves a big uncertainty. Therefore, slip rates are less helpful than it seems. Practically, we need the fault angle, the slope of the facet and that of the hillslopes in the transverse valleys. Surely possible to derive these properties from DEMs, but rather a student's thesis than a task for an afternoon.

*235. One mesh width? 60 km? Or one grid cell width, $\delta x$?*

"Mesh width" is quite usual in the context of finite differences, while "cell size" is typically used in the context of finite volumes. Anyway, **I replaced it by "grid spacing" (lines 259–260).**

*236. Plain → plane*

**Fixed (line 260),** thanks!

*237. Evaluating in the middle of the displacement steps: did you do a sensitivity test on this? Why not at the end of the step when more time for adjustment has occurred since the last step?*

I indeed started with a version that considered the topography at the end of the step, so immediately before the next displacement was applied. Then I followed theoretical arguments about differential equations (e.g., about source terms) that the middle of the interval should be better. I found that the difference is very small, but stayed at this version.

*Fig. 7 caption needs to explain that the lines are the vertical map projections of the facets.*

Indeed, **I added it.**

*Figure 8 caption or legend needs to indicate that the shaded areas are now the facets corresponding to the same colors/times as the drainage lines.*

Indeed, **I added it.**

*Section 5.2. I guess this is useful to compare to earlier models, but do vertically-slipping vertical-dipping faults exist in nature?*

Maybe in calderas, but this is not the subject here. The purpose of this section is even not the comparison to earlier models, but finding out whether or not horizontal displacement is important for the formation of facets. **I added "although of minor geological relevance" (lines 271–272).**

*254. I'd be careful with the word "impossible" unless the scope is clearly defined. (i.e. under these model rules)*

I think it should be clear that the entire section describes the results of a specific simulation. Furthermore, I already wrote "seems to be impossible".

*271–272. This is hard to follow. The fault to the profile – what profile? Where are the knickpoints arriving for both rivers and hillslopes?*

I think the profile shown in Fig. 9 was the only profile considered here. **I tried to describe it in more detail (lines 299–307).**

*283–286. Is it the original trace of the fault that is important, or the total vertical uplift so far? For incision = uplift, only after the total uplift is equal to the eventual steady height of the facets/ridgelines is the form fully adjusted to the new state. (e.g. for SPIM, Whipple and Tucker, 1999; but also more generally)*

It is indeed the original fault trace here, which separates the region with an inherited drainage pattern from a pristine surface. It is responsible for the conversion of multiangular facets into triangular facets. In turn, the vertical equilibration is responsible for the planarity of the facets, which is achieved faster. **I tried to clarify that the subsequent paragraph addresses this aspect and added some more explanation (lines 321–325).**

*294–298. How does the density of drainages at the range front under this model compare with e.g. Perron 2008 and its nonlinear-diffusion equivalent?*

Similar to the model used here, the linear diffusion model allows for adjusting the drainage density and the relief to any values, given that the scaling problems of the diffusion approach are not crucial. For the model used here, the horizontal length scale (that is inversely proportional to the drainage density) is $\frac{\kappa}{K}$, while it is $\sqrt{\frac{D}{K}}$ for the linear diffusion model. For the nonlinear diffusion model, however, I am not sure about its scaling properties.

*Fig 11, others – the 4My drainage networks are very disorganized and densified relative to typical network evolution simulations, is this an artifact of the channel-head identifying routine, or something dynamical related to normal faults and horizontal displacement?*

It is neither an immediate effect of normal faulting nor a clear artifact of the channel-head identifying scheme. **I extended the explanation in the context of Fig. 8 a bit (lines 282–288).**

*310–315, as before, if you have the same vertical uplift rate in the horst block between simulations, the time to equilibration of the form will also be similar.*

This is basically true, but I did not explain it here because I was sure that reviewers would complain that it is not explained sufficiently well. In principle, the results of my 2021 paper in JGR Earth Surface on knickpoints in the shared stream-power model can be transferred to hillslopes, but it remains more complicated than for the SPIM. In a nutshell, equlibration will take longer than in the SPIM, but the effect does not depend on the horizontal rate of displacement. I think it is not too bad that readers who are familiar with the SPIM may find this result not surprising even without an explcit explanation.

*335. What is the horizontal width, rather than the baseline? The facet length measured perpendicular to the range? This will be controlled by the height and angle, and therefore more related to the vertical motion, see previous comment.*

It was defined in line 184. **I added "normal to the baseline" (line 359).** However, it is not true that the horizontal width is controlled by the height and angle since everything can be rescaled vertically. So the horizontal width is the fundamental property beyond the baseline length.

*342. I → It*

**Fixed (line 377),** thanks!

*343. "Some kind of" → "a"*

**Changed (line 378).**

*Finally, I was left wondering: What would nonlinear-flux hillslope transport predict for the steady relief of the ridge lines between drainages (e.g. Roering 1999/2001) and how does this relate to facet height?*

I am not sure whether I got the question fully, but I guess it refers to the nonlinear diffusion model with $D \to \infty$ if the slope approaches a limit slope $S_c$. Then the facets and the hillslopes approach the same slope $S_f = S_h = S_c$ if the fault is steep enough and displacement fast enough. Except that the slopes of the facets and the hillslopes in the valleys woulb be practically the same, everything else could also be adjusted easily.

**Reviewer 2**

Unfortunately, I only found an overall assessment, but no comments.